

# Ideas and perspectives: Hydrothermally driven redistribution and sequestration of early Archaean biomass—the 'hydrothermal pump hypothesis'

Jan-Peter Duda[1,2,*], Volker Thiel[1], Thorsten Bauersachs[3], Helge Mißbach[1,4], Manuel Reinhardt[1,4], Nadine Schäfer[1,2], Martin J. Van Kranendonk[5,6,7], Joachim Reitner[1,2]

[1]Department of Geobiology, Geoscience Centre, Georg-August-University Göttingen, Goldschmidtstraße 3, 37077 Göttingen, Germany
[2]'Origin of Life' Group, Göttingen Academy of Sciences and Humanities, Theaterstraße 7, 37073 Göttingen, Germany
[3]Department of Organic Geochemistry, Institute of Geosciences, Christian-Albrechts-University Kiel, Ludewig-Meyn-Straße 10, 24118 Kiel, Germany
[4]Department Planets and Comets, Max Planck Institute for Solar System Research, Justus-von-Liebig-Weg 3, 37077 Göttingen, Germany
[5]Australian Centre for Astrobiology, University of New South Wales, Kensington, New South Wales 2052, Australia
[6]School of Biological, Earth and Environmental Sciences, University of New South Wales, Kensington, New South Wales 2052, Australia
[7]Australian Research Council Centre of Excellence for Core to Crust Fluid Systems, School of Biological, Earth and Environmental Sciences, University of New South Wales, Kensington, New South Wales 2052, Australia

*Correspondence to*: Jan-Peter Duda (jan-peter.duda@geo.uni-goettingen.de)

**Abstract.** Archaean hydrothermal chert veins commonly contain abundant organic carbon of uncertain origin (abiotic vs. biotic). In this study, we analysed kerogen contained in a hydrothermal chert vein from the ca. 3.5 Ga old Dresser Formation (Pilbara Craton, Western Australia). Catalytic hydropyrolysis (HyPy) of this kerogen yielded $n$-alkanes up to $n$-$C_{22}$, with a sharp decrease in abundance beyond $n$-$C_{18}$. A very similar distribution ($\leq n$-$C_{18}$) was observed in HyPy products of pre-extracted–recent bacterial biomass, while abiotic compounds synthesised via Fischer-Tropsch-type synthesis exhibited a modal distribution. We therefore propose that the original organic matter in the Archaean chert veins has a primarily microbial origin. We hypothesise that the microbially-derived organic matter accumulated in different aquatic and subsurface Dresser environments, and was then assimilated, redistributed and sequestered by hydrothermal fluids ('hydrothermal pump hypothesis').

## 1 Introduction

Extensive hydrothermal chert vein systems containing abundant organic carbon are a unique phenomenon of early Archaean successions worldwide (Lindsay et al., 2005; Van Kranendonk, 2006). A dense stockwork of several hundred kerogen-rich hydrothermal chert veins of the ca. 3.5 Ga Dresser Formation (Pilbara Craton, Western Australia; Fig. 1) are up to 2 km deep



by 25 m wide and penetrate footwall pillowed komatiitic basalts (Nijman et al., 1999; Van Kranendonk and Pirajno, 2004; Lindsay et al., 2005; Van Kranendonk, 2006; Van Kranendonk et al., 2008) (Fig. S1). Depleted stable carbon isotope signatures ($\delta^{13}C$) of bulk kerogens (–38.1 ‰ to –24.3 ‰) and of organic microstructures (–33.6 ‰ to –25.7 ‰) in these hydrothermal chert veins are consistent with a biological origin of the organic matter (Ueno et al., 2001, 2004; Glikson et al.,

2008; Pinti et al., 2009; Morag et al., 2016). Problematically, however, similarly depleted $\delta^{13}C$ values (partly down to ca. –36 ‰ relative to the initial substrate) can also be formed through abiotic processes, for instance via Fischer-Tropsch-type synthesis (McCollom et al., 1999; McCollom and Seewald, 2006).

Organic biomarkers can help to trace life and biological processes through deep time and add important information on the origin of the organic matter even in very old sedimentary rocks (Brocks and Summons, 2003; Summons and Hallmann,

2014). In Archaean rocks, however, molecular fingerprints in the conventionally analysed bitumen (i.e. the extractable portion of organic matter) are commonly blurred by thermal maturation and/or contamination (Brocks et al., 2008; Brocks, 2011; French et al., 2015). In contrast, the non-extractable portion of organic matter, known as kerogen, tends to be less affected by thermal maturation and contamination and is considered to be syngenetic with the host rock (Love et al., 1995; Brocks et al., 2003b; Marshall et al., 2007; Lockhart et al., 2008). Catalytic hydropyrolysis (HyPy) is a powerful tool for

sensitively releasing kerogen-bound hydrocarbon moieties with little structural alteration (Love et al., 1995). HyPy has been successfully applied to Archaean kerogens in rocks from the Pilbara Craton, liberating syngenetic organic compounds consistent with a biogenic origin (Brocks et al., 2003b; Marshall et al., 2007). However, this technique has not been used on kerogens contained in hydrothermal chert veins of this age yet.

Here, we present the results of analyses of kerogen embedded in a freshly exposed hydrothermal chert vein of the ca. 3.5 Ga

Dresser Formation. Our analyses include field and petrographic observations, Raman spectroscopy, and organic geochemistry ($\delta^{13}C_{TOC}$; kerogen-bound molecules via HyPy followed by gas chromatography-mass spectrometry (GC-MS) and gas chromatography-combustion-isotope ratio mass spectrometry (GC-C-IRMS)). To further constrain potential sources of the Dresser kerogen, we additionally applied HyPy on (i) excessively pre-extracted cyanobacterial biomass and (ii) produced abiotic organic matter via Fischer-Tropsch-type synthesis using a hydrothermal reactor. Results of these

investigations suggest that the Dresser kerogen has a primarily microbial origin. We hypothesise that biomass-derived organic compounds accumulated in different Dresser environments and were then redistributed and sequestered by subsurface hydrothermal fluids ('hydrothermal pump hypothesis').

## 2 Material & methods

### 2.1 Sample preparation

A fresh decimetre-sized sample of a Dresser chert vein was obtained from a recent cut wall of the abandoned Dresser Mine in the Pilbara Craton, Western Australia (Figs. 1, S1d). External surfaces (ca. 1–2 cm) of the sample block were removed using an acetone-cleaned rock saw and then used for the preparation of thin sections. The surfaces of the resulting inner





block were extensively rinsed with acetone and then removed (ca. 1–2 cm) with an acetone-cleaned high precision saw (Buehler; Isomet 1000, Germany). The surfaces of the resulting sample were again rinsed with acetone, and then crushed and powdered using a carefully acetone-cleaned pebble mill (Retsch MM 301, Germany).

## 2.2 Petrography and Raman spectroscopy

Petrographic analysis was conducted using a Zeiss SteREO Discovery.V8 stereomicroscope (transmitted and reflected light) linked to an AxioCam MRc5 5-megapixel camera.

Raman spectra were recorded using a Horiba Jobin Yvon LabRam-HR 800 UV spectrometer (focal length of 800 mm) attached to an Olympus BX41 microscope. For excitation an Argon ion laser (Melles Griot IMA 106020B0S) with a laser strength of 20 mW was used. The laser beam was focused onto the sample using an Olympus MPlane 100 x objective with a numerical aperture of 0.9 and dispersed by a 600 l mm$^{-1}$ grating on a CCD detector with 1024 x 256 pixels. This yielded a spectral resolution of <2 cm$^{-1}$ per pixel. Data were acquired over 10 to 30 s for a spectral range of 100–4000 cm$^{-1}$. The spectrometer was calibrated by using a silicon standard with a major peak at 520.4 cm$^{-1}$. All spectra were recorded and processed using the LabSpec$^{TM}$ database (version 5.19.17; Jobin-Yvon, Villeneuve d'Ascq, France).

## 2.3 Raman-derived H/C data

H/C values were calculated based on integrated peak intensities of Raman spectra using the formula H/C = 0.871 * $I_{D5}/(I_{G+D2})$ - 0.0508 (Ferralis et al., 2016).

## 2.4 Molecular analysis of the Dresser kerogen

All materials used for preparation were heated to 500°C for 3h and/or extensively rinsed with acetone prior to sample contact. A laboratory blank was prepared and analysed in parallel to monitor laboratory contaminations.

We applied catalytic hydropyrolysis (HyPy) to release molecules from the Dresser kerogen and pre-extracted biomass of the heterocystous cyanobacterium *Anabaena cylindrica* following existing protocols (Snape et al., 1989; Love et al., 1995; Love et al., 2005). HyPy allows the breaking of covalent bonds by progressive heating under high hydrogen pressure (150 bar). The released products are immediately removed from the hot zone by a constant hydrogen flow, and trapped downstream on clean, combusted silica powder cooled with dry ice (Meredith et al., 2004). All hydropyrolysates were eluted from the silica trap with high-purity dichloromethane (DCM), desulfurised overnight using activated copper, and subjected to gas chromatography-mass spectrometry (GC-MS).Before all experiments, the empty HyPy system was heated (ambient temperature to 520°C, held for 30min) to remove any residual molecules.

Isolation of the Dresser kerogen followed standard procedures (cf. Brocks et al., 2003b). 57 g of powdered sample was first demineralised with hydrochloric acid (24h) and then hydrofluoric acid (11 days). The purified organic matter was then




exhaustively extracted using 3 excess volumes of DCM and *n*-hexane, respectively (x3), ultrasonic swelling in pyridine (2 x 20 min at 80°C), and ultrasonic extraction with DCM (x3) and DCM/methanol (1/1; v/v). The kerogen was subsequently extracted with *n*-hexane until no more compounds were detected via GC-MS. The pure kerogen was used for HyPy.

In order to monitor potential contamination, we applied HyPy to blanks before and after the kerogen run. The Dresser

kerogen (131.06 mg) and blanks were sequentially heated in the presence of a sulphided molybdenum catalyst (10 wt%) and under a constant hydrogen flow (5 $dm^3$ $min^{-1}$) using a two-step approach (cf. Brocks et al., 2003b; Marshall et al., 2007; Fig. S2). The low temperature step included heating from ambient temperature to 250°C (300°C $min^{-1}$) and then to 330°C (8°C $min^{-1}$) to release any molecules that were strongly adsorbed to the kerogen and not accessible to solvent extraction. The silica powder was subsequently recovered from the product trap, and the trap was refilled with clean, combusted silica powder for

the following high temperature step. This step included heating from ambient temperature to 520°C (8°C $min^{-1}$) to release molecules covalently bound to the kerogen.

## 2.5 Molecular analysis of pre-extracted cyanobacterial biomass (*Anabaena cylindrica*)

We used cell material of the heterocystous cyanobacterium *Anabaena cylindrica* strain SAG 1403-2 as it fulfils all criteria for reference material (availability, well characterized, etc.). The material was obtained from the Culture Collection of Algae

at the Georg-August-University Göttingen, Germany. The axenic batch culture was grown in 250 ml of $BG11_0$ medium free of combined nitrogen sources (Rippka et al., 1979) at 29°C. A light/dark regime of 14 h:10 h with a photon flux density of 135 μmol photons $m^{-2}s^{-1}$ was provided by a white fluorescent light bulb. At the end of the logarithmic growth phase, cells were harvested by centrifugation and they were lyophilised thereafter. Subsequently, an aliquot of the freeze-dried biomass was exhaustively extracted following the methodology described by ref. (Bauersachs et al., 2014).

HyPy on the extraction residue was performed according to Love et al. (2005). The material was heated in the presence of a sulphided molybdenum catalyst (1.5 times weight of the bacterial extraction residue) and under a constant hydrogen flow (6 $dm^3$ $min^{-1}$). The HyPy program included heating from ambient temperature to 260°C (300°C $min^{-1}$) and then to 500°C (8°C $min^{-1}$).

## 2.6 Fischer-Tropsch-type synthesis of organic matter under hydrothermal conditions

Fischer-Tropsch-type reactions were carried out based on McCollom et al. (1999). A mixture of 2.5 g oxalic acid (dihydrate, suprapur®, Merck KGaA), 1 g montmorillonite (K10, Sigma Aldrich; pre-extracted with DCM; x4) and ca. 11 ml ultrapure water was heated to 175°C for 66 to 74 h in a sealed Morey-type stainless steel autoclave. The autoclave was rapidly (≤ 15 min) cooled to room temperature with compressed air.

Fluid- and solid phases were collected and extracted with DCM (x4). The montmorillonite was removed by centrifugation

and filtration with silica powder and sea sand (combusted, respectively). The obtained extracts were then concentrated by rotary evaporation (40°C, 670 mbar) and subjected to GC-MS analysis. Analytical blank experiments were carried out with all reactants to keep track of contamination.





## 2.7 Gas chromatography-mass spectrometry (GC-MS)

GC-MS analysis of HyPy products was carried out with a Thermo Scientific Trace 1300 Series GC coupled to a Thermo Scientific Quantum XLS Ultra MS. The GC instrument was equipped with a capillary column (Phenomenex Zebron ZB-5, 30 m, 0.1 µm film thickness, 0.25 mm inner diameter). Fractions were injected into a splitless injector and transferred to the GC column at 270°C. He was used as carrier gas with a constant flow rate of 1.5 ml min$^{-1}$. The GC oven temperature was held isothermal at 80°C for 1 min and then ramped to 310°C at 5°C min$^{-1}$, at which it was kept for 20 min. Electron ionization mass spectra were recorded in full scan mode at an electron energy of 70 eV with a mass range of $m/z$ 50–600 and scan time of 0.42 s.

## 2.8 Polyaromatic hydrocarbon (PAH) ratio definitions

1. Methylnaphthalene ratio (MNR) = 2-MN/1-MN (Radke et al., 1984)
2. Methylphenanthrene index (MPI-I) = 1.5 * (2-MP + 3-MP)/(P + 1-MP + 9-MP) (Radke and Welte, 1983)
3. Computed vitrinite reflectance [Rc (MPI-I)] = 0.7 * MPI-1 + 0.22 (according to P/MP >1; Boreham et al., 1988)

MN = Methylnaphthalene; P = Phenanthrene; MP = Methylphenanthrene.

## 2.9 Total organic carbon (TOC) and δ13C analyses (TOC and compound specific)

TOC, $\delta^{13}C_{TOC}$, and compound specific $\delta^{13}C$ analyses were conducted at the Centre for Stable Isotope Research and Analysis (KOSI) at the Georg-August-University Göttingen, Germany. Stable carbon isotope data are expressed as delta values relative to the Vienna Pee Dee Belemnite (VPDB) reference standard.

The TOC content and $\delta^{13}C_{TOC}$ values were determined in duplicate using an elemental analyser (NA-2500 CE-Instruments) coupled to an isotope ratio mass spectrometer (Finnigan MAT Delta plus). Ca. 100 mg of powdered and homogenised whole rock material were analysed in each run. For internal calibration an acetanilide standard was used ($\delta^{13}C$ = –29.6 ‰; SD = 0.1). TOC content measurements showed a mean deviation of 0.1. The average $\delta^{13}C_{TOC}$ value had a standard deviation of 0.3. Compound specific $\delta^{13}C$ analyses were conducted with a Trace GC coupled to a Delta Plus isotope ratio mass spectrometer (IRMS) via a combustion-interface (all Thermo Scientific). The combustion reactor contained CuO, Ni and Pt and was operated at 940°C. The GC was equipped with two capillary columns (Agilent DB-5 and DB-1; each 30 m, 0.25 µm film thickness, 250 µm inner diameter). Fractions were injected into a splitless injector and transferred to the GC column at 290°C. The carrier gas was He at a flow rate of 1.2 ml min$^{-1}$. The GC oven temperature program was identical to the one used for GC-MS analysis (see above). $CO_2$ with known $\delta^{13}C$ value was used for internal calibration. Instrument precision was checked using a mixture of $n$-alkanes with known isotopic composition. Standard deviations of duplicate measurements were better than 1.7.



## 3 Results

The hydrothermal chert vein studied (GPS: 021°09'04.13" S; 119°26'15.21" E; Fig. 1) is hosted in komatiitic pillow basalts that have undergone severe hydrothermal acid-sulphate alteration, producing a kaolinite-illite-quartz mineral assemblage (Van Kranendonk, 2006; Van Kranendonk and Pirajno, 2004) (Fig. S1). The sampled vein crops out in a recent cut wall of

the abandoned Dresser Mine (Fig. S1) and consists of a dense chert (microquartz) matrix of deep black colour that contains local concentrations of fresh (unweathered) pyrite crystals (Fig. 2a, b).

Petrographic analysis and Raman-spectroscopy revealed that kerogen (D bands at 1353 cm$^{-1}$, G bands at 1602 cm$^{-1}$) is embedded in the chert matrix (SiO$_2$ band at 464 cm$^{-1}$) (see Fig. 2c for a representative Raman spectrum). The organic matter occurs as small clots (<50 μm) of variable shape. Raman spectra of the organic matter exhibit relatively wide D and G bands

(82 and 55 cm$^{-1}$ width, respectively) (Fig. 2c). The total organic carbon (TOC) content is 0.2 wt%, and the δ$^{13}$C$_{TOC}$ value is −32.8±0.3 ‰ (Tab. 1). Raman-derived H/C ratios (calculated after Ferralis et al., 2016) range between 0.03 and 0.14.

HyPy was applied to the isolated Dresser kerogen, as well as to preceding and subsequent analytical blanks (combusted sea sand). Hydropyrolysates of the preceding blank contained a series of $n$-alkanes ≤ $n$-C$_{24}$, with maximum abundances at $n$-C$_{15}$ (Figs. 3, S2–S4). Furthermore, sulphur (only after low temperature HyPy), traces of siloxanes, and a phenol (Figs. 3, S2, S4)

were present. The blank runs also contained traces of aromatic hydrocarbons (Fig. S5).

Low temperature HyPy of the Dresser kerogen produced traces of C$_{14–18}$ $n$-alkanes, with a maximum at $n$-C$_{15}$ (Fig. S3). However, these compounds were significantly less abundant than those released during high temperature HyPy (see below). Other compounds observed in the low temperature step pyrolysate were elemental sulphur, phenols, phthalic acid, siloxanes and traces of aromatic hydrocarbons (Figs. S2, S4–S5).

High temperature HyPy of the Dresser kerogen yielded $n$-alkanes ranging from $n$-C$_{11}$ to $n$-C$_{22}$ with a notable decrease ('step') in the abundance of homologues above $n$-C$_{18}$ (Figs. 3b, S2, S3), which remained virtually unaffected by blank subtraction ('step' in Fig. 3c). Apart from that 'step', no carbon number preference is evident. The $n$-alkanes have δ$^{13}$C values ranging from −29.4 ‰ to −33.3 ‰ (mean −31.4±1.2 ‰; Fig. 3b; Tab. 1). The hydropyrolysate also contained isomeric mixtures of C$_{12–18}$ monomethylalkanes (Fig. S3) and a variety of aromatic hydrocarbons, including (dimethyl-, methyl-)naphthalene(s),

(methyl-)biphenyl(s), (methyl-)acenaphthene(s), dibenzofuran, and (methyl-)phenanthrene(s) (Figs. 3b, c, S4–S6). Biologically-diagnostic hydrocarbons such hopanoids or steroids were absent.

HyPy-treatment of pre-extracted biomass of the heterocystous cyanobacterium *Anabaena cylindrica* SAG 1403-2 yielded a variety of organic compounds, but also included $n$-alkanes with a clear restriction in carbon number to homologues ≤ $n$-C$_{18}$ (Fig. 3d). In contrast, our experimental synthesis of abiotic $n$-alkanes through Fischer-Tropsch-type reactions under

hydrothermal conditions produced a modal distribution of homologues without any carbon number preference (Fig. 3e).



## 4 Discussion

### 4.1 Maturity of the Dresser kerogen

The organic record of Archaean rocks is commonly affected by thermal maturation (Brocks et al., 2008; Brocks, 2011; French et al., 2015). The vein chert studied here was obtained from a fresh outcrop and shows no evidence for weathering

(e.g., re-oxidation of pyrite; Fig. 2b). The kerogen within the sample is thermally mature and structurally disordered, as evidenced by relatively wide D and G bands in the Raman spectra (see Fig. 2c for a representative Raman spectrum). This is further supported by the absence of a S1 band at 2450 cm$^{-1}$, consistent with prehnite-pumpellyite and lower greenschist metamorphism (cf. Yui et al., 1996). Both observations are well in line with published Raman spectra of Archaean organic matter from the same region (Ueno et al., 2001; Marshall et al., 2007) and the general thermal history of the host rock

(regional prehnite–pumpellyite to lower greenschist metamorphism; Hickman, 1983; Terabayashi et al., 2003). The kerogen is embedded in a very dense and relatively impermeable chert matrix, similar to indigenous and syngenetic organic matter in other hydrothermal chert veins of the Dresser Formation (Morag et al., 2016). An introduction of younger solid macromolecular organic matter during later fluid flow phases, as proposed for the younger Apex chert (Olcott-Marshall et al., 2014), therefore appears highly unlikely.

The observed spread in the Raman-derived H/C ratios (0.03–0.14) is due to uncertainties in peak integration. However, both the H/C ratios and the methylphenanthrene/phenanthrene value (MP/P = 0.33; Tab. 2) of the Dresser kerogen are in good accordance with data reported from more mature Strelley Pool kerogens (0.08–0.14 and 0.37–0.24, respectively; Marshall et al., 2007). The low methylphenanthrene index (MPI-I = 0.23) and high phenanthrene/methylphenanthrene index (P/MP = 3.01) result in a computed vitrinite reflectance ($R_c$ (MPI-I)) of 2.87 (Tab. 2), indicating a thermal maturity far beyond the oil

generative stage (Radke and Welte, 1983; Boreham et al., 1988). However, it has to be considered that the MPI-I is potentially affected by (de-) methylation reactions (Brocks et al., 2003a). The calculated methylnaphthalene ratio (MNR = 2.52) corresponds to a somewhat lower mean vitrinite reflectance of ca. 1.5 (Tab. 2), which is again in line with a post oil window maturity (cf. Radke et al., 1984). The mismatches between single aromatic maturity parameters are negligible as these indices have limited application for highly mature Archaean organic matter (Brocks et al., 2003a). The putative offset

between the indices and the metamorphic overprint indicated by Raman data is most likely due to the protection of kerogen-bound moieties even under elevated thermal stress (Love et al., 1995; Lockhart et al., 2008). Furthermore, it has been shown that kerogen isolated from the ca. 3.4 Ga Strelley Pool Formation of the Pilbara Craton also contains larger PAH clusters which are not GC-amenable (>10-15 rings; Marshall et al., 2007). Consequently, it can be anticipated that the compounds detected in the HyPy pyrolysate of the Dresser kerogen represent only a small fraction of the bulk macromolecular organic

matter.

The distribution of monomethylalkanes ≤ $n$-C$_{18}$ (Figs. S3, S4) released from the Dresser kerogen resembles high temperature HyPy products from the Strelley Pool kerogen (Marshall et al., 2007; their Fig. 15). Such isomeric mixtures are typically formed during thermal cracking of alkyl moieties (Kissin, 1987) and are in good agreement with the estimated maturity



range. Methylated aromatics such as methylnaphthalenes and -phenanthrenes have also been observed in other hydropyrolysates from Archaean kerogens that experienced low regional metamorphism (French et al., 2015; Brocks et al., 2003b; Marshall et al., 2007). In all of these cases, the degree of alkylation varied with the exact thermal alteration of the respective kerogens (French et al., 2015; Marshall et al., 2007).

## 4.2 Syngeneity of the Dresser kerogen-derived compounds

As the bitumen fractions of Precambrian rocks are easily biased by the incorporation of contaminants during later stages of rock history (Brocks et al., 2008; Brocks, 2011; French et al., 2015), studies have increasingly focussed on kerogen-bound compounds that are more likely to be syngenetic to the host rock (Love et al., 1995; Brocks et al., 2003b; Marshall et al., 2007; French et al., 2015). Potential volatile organic contaminants adhering to the kerogen are removed through excessive extraction and a thermal desorption step (~350°C) prior to high temperature HyPy (550°C; cf. Brocks et al., 2003b; Marshall et al., 2007). The recurrence of few $n$-alkanes in the range of $n$-$C_{14}$ to $n$-$C_{24}$ (maximum at $n$-$C_{15}$) in high temperature HyPy blank runs obtained immediately before and after the actual sample run (Figs. 3a, S2, S3) indicates minor background contamination during pyrolysis, with a source most likely within the HyPy system. However, these contaminants do not significantly affect the $n$-alkane pattern yielded by high temperature HyPy of the Dresser kerogen, as evidenced by blank subtraction (Fig. 3c).

Contamination of the sample can be further deciphered by the presence of polar additives or hydrocarbons that are not consistent with the thermal history of the host rock. Plastic-derived branched alkanes with quaternary carbon centres (BAQCs), a common contaminant in Precambrian rock samples (Brocks et al., 2008), have not been detected (Fig. S7). Traces of functionalized plasticizers (phenols and phthalic acid; Figs. S2, S6, S8) are unlikely to survive (or result from) HyPy-treatment. These compounds are therefore considered as background contamination introduced during sample preparation and analysis after HyPy. The observed siloxanes (Fig. 3, S2) probably originate from the GC column or septum bleeding and are unlikely to be contained in the sample. All of these compounds occur only in low or trace abundances and can be clearly distinguished from the ancient aliphatic and aromatic hydrocarbons contained in the Dresser kerogen.

Contamination by (sub-)recent endoliths can be excluded as sample surfaces have been carefully removed and there is no petrographic indication for borings or fissures containing recent organic material (Fig. 2). HyPy of untreated or extracted biomass would yield a variety of acyclic and cyclic biomarkers (cf. Love et al., 2005). However, high temperature HyPy of the Dresser kerogen almost exclusively yielded $n$-alkanes, minor amounts of monomethylalkanes, and various aromatic hydrocarbons (Figs. 3b, c, S2–S6), while hopanoids or steroids were absent (Figs. S8, S9). This is also in good agreement with the maturity of the Dresser kerogen and previous HyPy studies of Archaean rocks (French et al., 2015; Brocks et al., 2003b; Marshall et al., 2007).

Accidental contamination of the kerogen by mono-, di- and triglycerides (e.g., dust, skin surface lipids) can also be ruled out as HyPy-treatment of these compounds typically results in $n$-alkane distributions with a distinct predominance of $n$-$C_{16}$ and $n$-$C_{18}$ homologues, corresponding to the $n$-$C_{16}$ and $n$-$C_{18}$ fatty acid precursors (Love et al., 2005; Craig et al., 2004; own





observations). At the same time, the observed distribution of kerogen-derived *n*-alkanes, with a distinct decrease beyond *n*-$C_{18}$ (Fig. 3b, c), is notably similar to high temperature HyPy products of kerogens isolated from the ca. 3.4 Ga Strelley Pool Formation of the Pilbara Craton (Marshall et al., 2007; their Fig. 14). Marshall and co-workers considered these *n*-alkanes unlikely to be contaminants as they were released only in the high temperature HyPy step. Furthermore, the stable carbon

isotopic composition of *n*-alkanes in the Dresser high temperature pyrolysate (–29.4 ‰ to –33.3 ‰; mean –31.4±1.2 ‰) is very similar to the $\delta^{13}C_{TOC}$ signature (–32.8±0.3 ‰; Figs. 3, S10; Tab. 1), indicating that these compounds were generated from the kerogen. Hence, we consider the compounds released from the Dresser kerogen during high temperature HyPy (Fig. 3b, c) as syngenetic.

High temperature HyPy of the Dresser kerogen yielded a variety of aromatic hydrocarbons, which are orders of magnitudes

lower or absent in all other pyrolysates (Figs. 3, S2, S4–S6). It also produced significantly higher amounts of *n*-alkanes than the low temperature step, and these further showed a distinct distribution pattern (i.e. a step ≤ *n*-$C_{18}$; Figs. 3, S2, S3). The lack of even low quantities of *n*-alkenes that typically accompany bond cleavage in kerogen pyrolysis was also observed by Marshall et al. (2007) in their HyPy analysis of cherts from the Strelley Pool Formation. As a possible explanation for the lack of *n*-alkenes, these authors suggested that the *n*-alkanes were not cracked from the kerogen but were rather trapped in

closed micropores until the organic host matrix was pyrolytically disrupted. Alternatively, however, unsaturated cleavage-products may have been immediately reduced during HyPy by the steadily available hydrogen and catalysts. Indeed, it has been demonstrated that double bonds in linear alkyl chains are efficiently hydrogenated during HyPy, even in case of pre-extracted microbial biomass (e.g. Love et al., 2005, our Fig. 3d). We consider both as plausible scenarios to explain the absence of *n*-alkenes in the Dresser chert hydropyrolysate.

**4.3 Origin of the Dresser kerogen: hydrothermal vs. biological origin**

The $\delta^{13}C_{TOC}$ value of –32.8±0.3 ‰ is consistent with carbon fixation by photo- or chemoautotrophs (cf. Schidlowski, 2001). However, organic compounds exhibiting similar $^{13}C$ depletions (partly down to ca. –36 ‰ relative to the initial substrate) could also be formed abiotically during serpentinisation of ultramafic rocks (Fischer-Tropsch-type synthesis; McCollom et al., 1999; McCollom and Seewald, 2006; Proskurowski et al., 2008). A further constraint on the origin of the Dresser

kerogen is provided by the distinct decrease in *n*-alkane abundance beyond *n*-$C_{18}$ observed in the high temperature HyPy pyrolysate (Fig. 3b, c). This distribution resembles HyPy products of pre-extracted recent cyanobacterial biomass, which also shows a very pronounced restriction in carbon number to homologues ≤ *n*-$C_{18}$ (Fig. 3d).

The pre-extracted cyanobacterial biomass and the abiotically produced *n*-alkanes studied as reference samples experienced no thermal maturation. It can nevertheless be expected that burial, and thus heating of, *Anabaena* biomass (Fig. 3d) would

initially liberate lower *n*-alkane homologues from their predominant *n*-alkyl moieties while, up to a certain point, retaining the distinct step at *n*-$C_{18}$. Experimental maturation of an immature kerogen revealed the preservation of distinct alkyl-chain length preferences even after 100 days at 300°C (Mißbach et al., 2016; e.g. step beyond *n*-$C_{31}$ in their Fig. 2). Further maturation would ultimately lead to a modal distribution of short-chain *n*-alkanes and erase the biologically inherited pattern





(cf. Mißbach et al., 2016). In contrast, abiotically synthesised organic matter shows a modal homologue distribution from the beginning (Fig. 3e) and will retain it, while thermal maturation would gradually shift the *n*-alkane pattern towards shorter homologues. Consequently, the distinctive distribution of *n*-alkanes released from the Dresser kerogen (i.e. the step $\leq$ *n*-$C_{18}$; Figs. 3b, c) can be regarded as a molecular fingerprint relating to a biosynthetic origin of the organic matter.

Highly $^{13}C$ depleted methane in primary fluid inclusions in hydrothermal chert veins of the Dresser Formation ($\delta^{13}C$ <56 ‰) was taken as evidence for biological methanogenesis and thus the presence of Archaea (Ueno et al., 2006). Our $\delta^{13}C_{TOC}$ value (–32.8±0.3 ‰; Tab. 1; Fig. S10) would generally be consistent with both bacterial and archaeal sources (cf. Schidlowski, 2001). Straight-chain (acetyl-based) hydrocarbon moieties - the potential precursors of the kerogen-derived *n*-alkanes - are currently being formed only by Bacteria and Eukarya, whereas Archaea synthesize isoprene-based compounds

(Koga and Morii, 2007; Matsumi et al., 2011). Moreover, Bacteria commonly (though not exclusively) form linear carbon chains $\leq C_{18}$ (cf. Kaneda, 1991). Hence, given the lack of convincing evidence for the presence of Eukarya as early as 3.5 Ga (cf. French et al., 2015; Parfrey et al., 2011; Knoll, 2014), the most plausible source of kerogen-occluded *n*-alkanes in the Dresser hydrothermal chert is Bacteria. Given the largely anoxic conditions during deposition of the Dresser Formation (Van Kranendonk et al., 2003; Li et al., 2013), potential biological sources for the Dresser kerogen could include anoxygenic

photoautotrophic, chemoautotrophic, and heterotrophic microorganisms.

### 4.4 The 'hydrothermal pump hypothesis'

Our results strongly support a biological origin of the kerogen in the early Archaean hydrothermal chert veins of the Dresser Formation. We explain this finding by the redistribution and sequestration of microbial organic matter formed in different Dresser environments (see below) through hydrothermal circulation ('hydrothermal pump hypothesis'; Fig. 4). Higher

geothermal gradients prior to the onset of modern-type plate tectonics ($\geq$ 3.2–3.0 Ga; Smithies et al., 2005; Shirey and Richardson, 2011) were possibly important drivers of early Archaean hydrothermal systems. In fact, the Dresser Formation was formed in a volcanic caldera environment affected by strong hydrothermal circulation, where voluminous fluid circulation locally caused intense acid-sulphate alteration of basalts and the formation of a dense hydrothermal vein swarm (Nijman et al., 1999; Van Kranendonk and Pirajno, 2004; Van Kranendonk, 2006; Van Kranendonk et al., 2008; Harris et al.,

2009). In addition to the high crustal heat flow, the absence of thick sedimentary covers may have facilitated the intrusion of sea water into the hydrothermal system. The associated large-scale assimilation of particulate and dissolved organic matter and its transport and alteration by hydrothermal fluids (Fig. 4) therefore appears a plausible mechanism that may, at least partly, explain the high amounts of kerogen in early Archaean hydrothermal veins.

The 'hydrothermal pump hypothesis' requires a source of organic matter during the deposition of the Dresser Formation (Fig.

4). Whereas contributions from extra-terrestrial sources, as well as Fischer-Tropsch-type synthesis linked to serpentinisation of ultramafic rocks cannot be excluded, our results indicate a primarily biological origin for the kerogen contained in the chert veins (Fig. 4). The inferred biogenicity is also in line with the consistent $\delta^{13}C$ offset between bulk kerogens (ca. –20 to –30 ‰) and carbonate (ca. ±2 ‰) in Archaean rocks (Schidlowski, 2001; Hayes, 1983). Prokaryotic primary producers and



heterotrophs may have flourished in microbial mats (Dresser stromatolites; Walter et al., 1980; Van Kranendonk, 2006, 2011; Philippot et al., 2007; Van Kranendonk et al., 2008), the water column (planktic 'marine snow', Brasier et al., 2006; Blake et al., 2010), and hotsprings on land (Djokic et al., 2017). Another biological source for the ancient organic matter could have been chemoautotrophs and heterotrophs thriving in more cryptic environments such as basalts (Banerjee et al., 2007; Furnes

et al., 2008) and hydrothermal vent systems (Shen et al., 2001; Ueno et al., 2001, 2004, 2006; Pinti et al., 2009; Morag et al., 2016) (Fig. 4). All of these systems are not mutually exclusive and the largely anoxic conditions would have encouraged a high steady-state abundance of organic matter in the aquatic environment (Fig. 4).

Dissolved organic matter (DOM) in modern seawater may resist decomposition over millennial timescales (Druffel and Griffin, 2015). In recent hydrothermal fields, however, organic matter becomes thermally altered and redistributed (Simoneit,

1993; Delacour et al., 2008; Konn et al., 2009). Laboratory experiments using marine DOM indicate that thermal alteration already occurs at temperatures >68–100°C, and efficient removal of organic molecules at 212–401°C (Hawkes et al., 2015, 2016). It has been argued, however, that such DOM removal may also be due to transformation into immiscible material through, for example, condensation (Castello et al., 2014) and/or defunctionalisation reactions (Hawkes et al., 2016). These processes, however, are as yet poorly understood. In the Dresser Formation, hydrothermal temperatures ranged from ca.

300°C at depth to 120°C near the palaeosurface, causing propylitic (ca. 250–350°C) and argillic alterations (ca. 100–200°C) of the host rocks (Van Kranendonk and Pirajno, 2004; Van Kranendonk et al., 2008; Harris et al., 2009). Given this variety of thermal regimes, and the generally anoxic nature of early Archaean sea water, it is likely that some of the organic substances underwent *in situ* alteration but no complete oxidation during fast hydrothermal circulation. The entrained organics would have been trapped in the chert that instantaneous precipitated from the ascending hydrothermal fluids due to

sub-surface cooling (cf. Van Kranendonk, 2006).

In summary, the 'hydrothermal pump hypothesis' (Fig. 4) includes (i) a net build-up of organic matter in different Dresser environments under largely anoxic conditions, (ii) a large-scale assimilation of particulate and dissolved organic matter from various biological sources and its subsurface transport and alteration by hydrothermal fluids, as well as (iii) its sequestration within hydrothermal chert veins as kerogen. This model explains the presence of abundant organic carbon in early Archaean

hydrothermal veins, as well as its morphological, structural and isotopic variability observed in the Dresser hydrothermal chert veins (Ueno et al., 2001, 2004; Pinti et al., 2009; Morag et al., 2016).

## 4 Conclusions

Kerogen embedded in a hydrothermal chert from the ca. 3.5 Ga old Dresser Formation (Pilbara Craton, Western Australia) is syngenetic. A biological origin is inferred from the presence of short-chain *n*-alkanes in high temperature HyPy pyrolysates,

showing a sharp decrease in homologue abundance beyond *n*-C$_{18}$. HyPy products of pre-extracted recent bacterial biomass exhibited a similar restriction to carbon chain-lengths ≤ *n*-C$_{18}$, while abiotically synthesised compounds exhibited a modal distribution. This interpretation is further consistent with the $\delta^{13}C_{TOC}$ value (–32.8±0.3 ‰) and the stable carbon isotopic

composition of *n*-alkanes in the Dresser high temperature pyrolysate (–29.4 ‰ to –33.3 ‰; mean –31.4±1.2 ‰). Based on these observations, we propose that the original organic matter was primarily biologically synthesised. We hypothesise that microbially-derived organic matter accumulating in the anoxic aquatic and subsurface environment was assimilated, redistributed and sequestered by hydrothermal fluids ('hydrothermal pump hypothesis').

**Supplement**

**Author contributions**

JPD, JR and MVK conducted the field work and designed the study. JR conducted petrographic analyses. NS performed Raman spectroscopic analyses. MR and JPD conducted pyrolysis experiments. HM conducted Fischer-Tropsch-type synthesis. TB prepared cyanobacterial cell material. JPD, HM, MR and VT performed biomarker analyses. JPD wrote the
manuscript. All authors discussed the results and provided input to the manuscript.

**Competing interests**

The authors declare that they have no competing financial interests.

**Acknowledgments**

This work was financially supported by the Deutsche Forschungsgemeinschaft (grant Du 1450/3-1, DFG Priority
Programme 1833 „Building a Habitable Earth", to JPD and JR; grant Th 713/11-1 to VT), the Courant Research Centre of the Georg-August-University Göttingen (DFG, German Excellence Program), the Göttingen Academy of Sciences and Humanities (to JPD and JR), the International Max Planck Research School for Solar System Science at the Georg-August-University Göttingen (to MR and HM), and the ARC Centre of Excellence for Core to Crust Fluid Systems (MVK). We thank M. Blumenberg, C. Conradt, W. Dröse, J. Dyckmans, A. Hackmann, M. Öztoprak and B.C. Schmidt for scientific and
technical support. J. Rochelmeier is thanked for assistance during sample extraction of *A. cylindrica*. This is publication number [X] of the Early Life Research Group (Department of Geobiology, Georg-August-University Göttingen; Göttingen Academy of Sciences and Humanities) and contribution [X] from the ARC Centre of Excellence for Core to Crust Fluid Systems.

We acknowledge support by the German Research Foundation and the Open Access Publication Funds of the Göttingen
University.



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

**Figures**

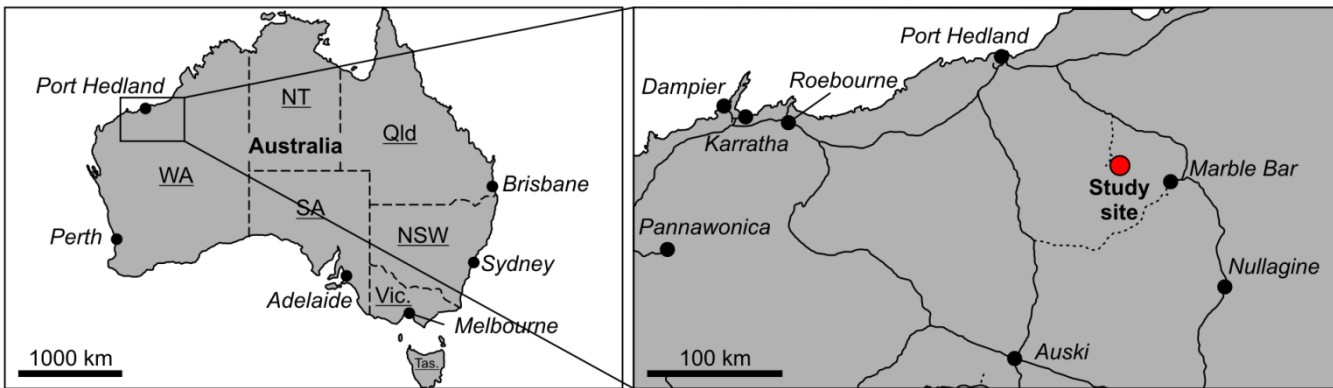

**Figure 1.** Location of the study site in Western Australia. The hydrothermal chert vein analysed occurs in a recent cut wall

5  of the abandoned Dresser Mine close to Marble Bar.




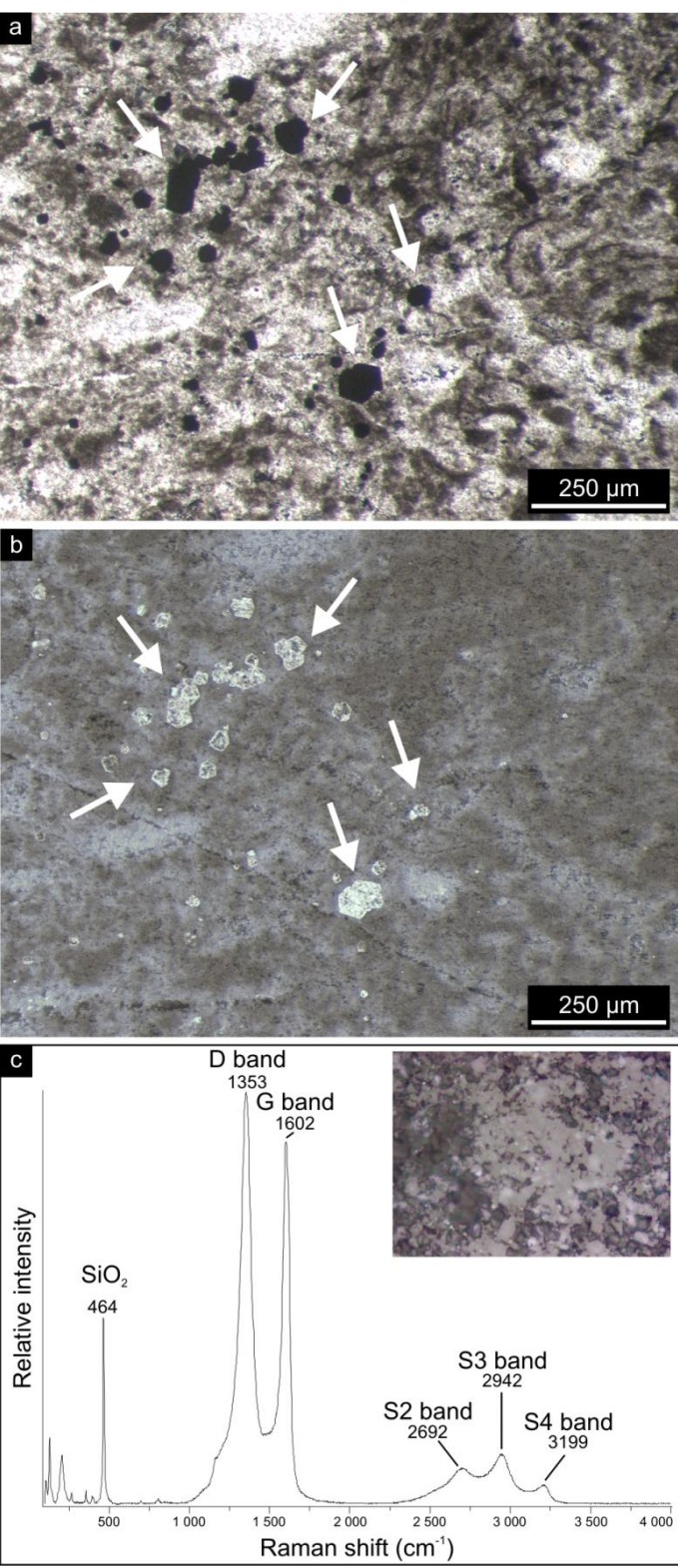



**Figure 2.** Petrographic observations on the hydrothermal chert vein. (**a,b**) Thin section photographs (**a:** transmitted light; **b:** reflected light) showing kerogen (brownish colours) and pyrite (arrows, black colours in **a**, bright colours in **b**) embedded within a fine grained chert matrix. Note that the pyrite crystals (arrows) are excellently preserved and show no evidence of oxidation. (**c**) Representative Raman spectrum of kerogen (D band at 1353 cm$^{-1}$ and G band at 1602 cm$^{-1}$) present in the hydrothermal chert vein. Note the wide D and G bands (82 and 55 cm$^{-1}$ width, respectively), pointing to thermally mature and structurally disordered kerogen, and the absence of a S1 band (at 2450 cm$^{-1}$) being consistent with prehnite-pumpellyite and lower greenschist metamorphosis (cf. Yui et al., 1996).







**Figure 3.** Total ion currents (a–c; in the same scale) and ion chromatograms selective for alkanes (d–f; *m/z* 85). (**a**) High temperature HyPy (330–520°C) product of analytical blank (combusted sea sand) obtained prior to HyPy of the Dresser kerogen. *n*-Alkanes in the range of *n*-C$_{14}$ to *n*-C$_{24}$ (maxima at *n*-C$_{15}$) represent HyPy background contamination. (**b**) High temperature HyPy (330–520°C) product of the Dresser kerogen. Note the sharp decrease in abundance of *n*-alkanes beyond

5   *n*-C$_{18}$ (see arrows). δ$^{13}$C values of *n*-C$_{12}$ to *n*-C$_{28}$ show a high similarity to the δ$^{13}$C$_{TOC}$ value (–32.8±0.3 ‰), further confirming syngeneity. (**c**) Blank subtraction (**b** minus **a**) showing that contaminants have no major impact on the *n*-alkane pattern yielded during the high temperature HyPy step of the Dresser kerogen (**b**). (**d**) HyPy products of cell material of the heterocystous cyanobacterium *Anabaena cylindrica*. Note the sharp decrease in abundance of *n*-alkanes beyond *n*-C$_{18}$ (see arrows), similar to the *n*-alkane distribution of the Dresser kerogen (**b**). (**e**) Products of experimental Fischer-Tropsch-type

10  synthesis under hydrothermal conditions; abiogenic *n*-alkanes show a modal distribution that is distinctly different from the Dresser kerogen. Black dots: *n*-alkanes (numbers refer to carbon chain lengths); N: naphthalene; MN: methylnaphthalenes; BiPh: 1,1'-biphenyl; DMN: dimethylnaphthalenes; DBF: dibenzofuran; MAN: methylacenaphthene; P: phenanthrene; crosses: siloxanes (GC column or septum bleeding).





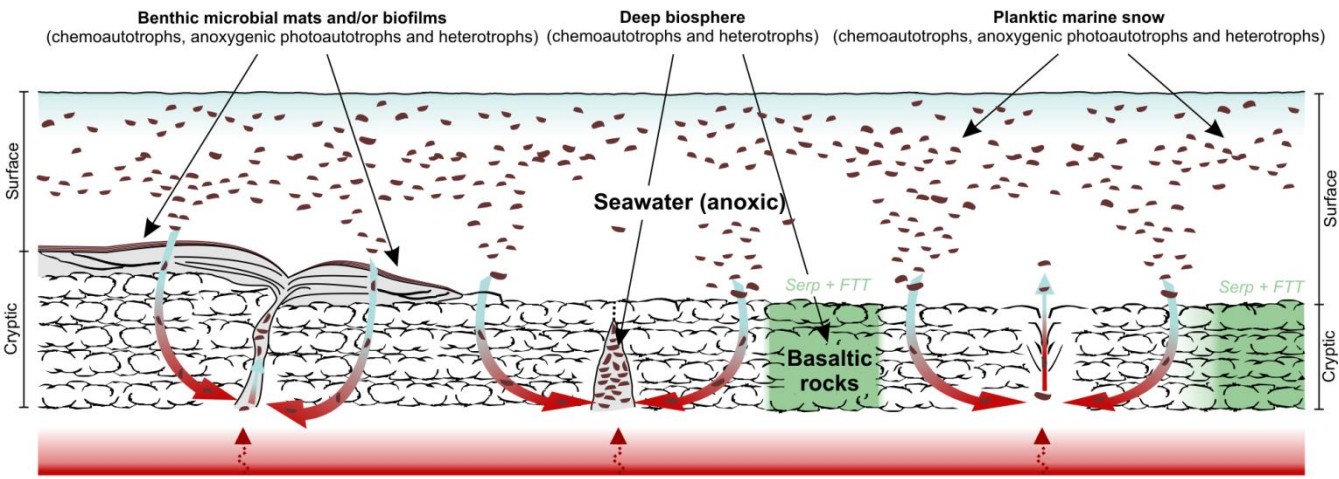

**Figure 4.** The 'hydrothermal pump hypothesis'. Organic matter was predominantly biologically produced and heterotrophically processed by Bacteria and, possibly, Archaea. Additionally, Fischer-Tropsch-type synthesis of organic matter linked to the serpentinisation of ultramafic rocks (McCollom et al., 1999; McCollom and Seewald, 2006) may have occurred locally. Primary producers (chemoautotrophs, anoxygenic photoautotrophs) and heterotrophs may have flourished in surface waters (planktic 'marine snow'), at the water/rock interface (microbial mats and/or biofilms), and in cryptic environments (e.g., within basalts and hydrothermal vent systems). After accumulating in different Dresser environments, the organic matter was redistributed and sequestered in veins by hydrothermal fluids.





**Table 1.** Stable carbon isotopic composition ($\delta^{13}$C) of the total organic carbon (TOC) and *n*-alkanes released from high temperature HyPy of the Dresser kerogen.

| | *n*-C$_{12}$ | *n*-C$_{13}$ | *n*-C$_{14}$ | *n*-C$_{15}$ | *n*-C$_{16}$ | *n*-C$_{17}$ | *n*-C$_{18}$ | *n*-C$_{12-18}$ (mean) | TOC |
|---|---|---|---|---|---|---|---|---|---|
| $\delta^{13}$C | -30.3 | -33.3 | -32.7 | -31.1 | -29.4 | -31.2 | -31.7 | -31.4 | -32.8 |
| SD | 0.5 | 1.4 | 0.2 | 1.7 | 0.3 | 0.8 | 0.1 | 1.2 | 0.3 |



**Table 2.** Maturity indices (based on aromatic hydrocarbons) of the Dresser kerogen. MP/P: Methylphenathrene/phenanthrene ratio; MPI-I: Methylphenanthrene index [1.5 * (2-MP + 3-MP)/(P + 1-MP + 9-MP); Radke and Welte, 1983]; P/MP: Phenanthrene/methylphenanthrene ratio; $R_c$ (MPI-I): computed vitrinite reflectance (0.7 * MPI-1 + 0.22, according to P/MP >1; Boreham et al. 1988); MNR: Methylnaphthalene ratio (2-MN/1-MN; Radke et al., 1984).

| MP/P | MPI-I | P/MP | $R_c$ (MPI-I) | MNR |
|------|-------|------|---------------|-----|
| 0.33 | 0.23 | 3.01 | 2.87 | 2.52 |