# Peer review of "Ideas and perspectives: Hydrothermally driven redistribution and sequestration of early Archaean biomass—the 'hydrothermal pump hypothesis'"

_Biogeosciences, 2017_

## Referee Comment (RC1) · M. Walter (Referee) · 21 Dec 2017

This is innovative, thorough and significant work addressing the biogenicity of the oldest known well preserved organic matter in the geological record and the biological affinities of its precursor organisms. The techniques involved have been applied rigorously but the interpretations are only as good as those techiques allow; I lack the expertise to judge them carefully so it is essential that this be done by well informed experts such as Roger Summons, Simon George and Jochen Brocks.

[Figure]

There is a need to discuss the possibility that there could have been contamination by organic compounds derived from the 2.7-2.8Ga Fortescue Group that overlies the Dresser Fm in the studied area. It is conceivable that a "hydrothermal pump" could have circulated fluids downwards into the older succession. There is a significant literature on the organic geochemistry of the Fortescue Group by George, Coffey, Summons and others. The same applies to the Strelley Pool Fm where there are abundant stromatolites and microfossils.

The manuscript would be enhanced by adding a paragraph outlining the evidence that indicates that the 3.5Ga environment was anoxic.

It seems to me that the evidence from the Apex Chert needs to be at least briefly reviewed here as it would add significantly to the context of this new work.

The manuscript is well written and almost free of errors. P.11 l. 19 change to instantaneously.

References to the published geological maps of the North Pole area should be added. There may be other publications by Hickman that should be cited.

---

## Referee Comment (RC2) · M.A. van Zuilen (Referee) · 7 Jan 2018

This is a very interesting paper showing a potential key characteristic of kerogen in chert veins of the early Archean Dresser Fm. The origin of the kerogen in this chert vein (and other early Archean chert veins in the Pilbara) still remains debated. It could represent sedimentary biomass that was recirculated by hydrothermal fluids, or represent the abiogenic product of FTT synthesis that was generated during hydrothermal serpentinization of ultramafic crust. This paper therefore addresses a very relevant scientific question.

Based on HyPy extraction, GC-MS and compound-specific isotope analysis, the authors showed that the kerogen phase releases n-alkanes with a specific carbon-number distribution. The sharp decrease in n-alkane abundance above n-C18, is similar to the distribution of n-alkanes released during HyPy of cyanobacteria. This distribution is not observed, however, in n-alkane products of FTT synthesis. Based on these observations, the authors conclude that the kerogen in the chert vein of the Dresser Fm is derived from biomass (such as e.g. phototrophs, chemolithoautrophs), and not from FTT synthesis associated with hydrothermal circulation. In order to explain the presence of kerogen in deep feeder chert veins, the authors propose a 'hydrothermal pump' hypothesis, in which redistribution and sequestration of microbial biomass occurs through hydrothermal circulation. This is a very nice explanation, and would confirm that microbial life was abundantly present in the ancient oceans.

The paper is very well written, and experimental results support the conclusions. I have 3 issues that should be worked out better in this paper, which are listed below.

1) The conclusion of this paper strongly depends on the drop in >C18 n-alkanes in biologic materials. It should then be explained in detail why this happens. Is this drop observed in all biologic materials? Is it related to specific compounds that are present in cell membranes? There is now only a very short description about this (P10, line 10-11), stating that bacteria commonly (though not exclusively) form linear carbon chains <C18 (cf. Kaneda, 1991). I think it is important that this discussion is expanded here. Is there a chemical reason for this drop in abundance of n-alkanes beyond C18?

2) The HyPy products of kerogen of the Dresser Fm are compared with HyPy products of cyanobacteria, but – if I understand correctly – not with the HyPy products of FTT synthesized carbonaceous solids. Only the direct gaseous FTT synthesis products are compared. Were there any solid phases produced during FTT synthesis? It would have been important to see what kind of HyPy products this would have generated, in comparison with that of HyPy products of cyanobacteria and Dresser Fm kerogen. I think the authors should discuss this better, in order to make clear that a comparison

with a true abiogenic carbonaceous solid has not been made here.

3) The maturity of kerogen in the Dresser Fm is determined using Raman spectroscopy. I have several comments to the methods used here. The H/C ratio of the kerogen fractions is calculated following the method of Ferralis et al. (2016), using a D5-peak in the Raman spectrum. However, based on the example spectrum in Fig.2c I think it is quite difficult to convincingly fit a D5-peak. The authors should at least show how this fit was made. It may well be that the kerogen has reached a degree of alteration where the amount of H is too low to create a significant D5-peak. The other method used here is to check the S1 peak (2450 cm-1) in the second order spectrum. The absence of this peak is consistent with kerogen that has experienced prehnite-pumpellyite to lower-greenschist-facies metamorphism. However, I think there are more precise methods to determine the degree of alteration of the kerogen. For instance, the Raman-based geothermometer of Koketsu et al. (2014) could have been applied to the first-order spectrum. Or even better, the Raman-based indicator of Delarue et al. (2016) can be applied. This indicator has been specifically developed to compare kerogens in Precambrian cherts.

Minor, technical issues:

- P3, Line 27: . . .(GC-MS). Before. . ..

- P4, Line 7 and line 22: Why is the heating program for the cyanobacteria different than for the extracted kerogens?

- Section 2.8: It may be good for the reader if a little more information is given here on PAH's. What are they, and why are these ratios important? And how were these PAH's measured?

- Section 4.4: The way it is written, it is not clear whether the 'hydrothermal pump hypothesis' is an already existing term, or is here proposed for the first time. It should be more clearly stated that this is a new proposed model.

---

## Author Comment (AC1) · 5 Feb 2018

Comment from referee: "This is innovative, thorough and significant work addressing the biogenicity of the oldest known well preserved organic matter in the geological record and the biological affinities of its precursor organisms. The techniques involved have been applied rigorously but the interpretations are only as good as those techiques allow; I lack the expertise to judge them carefully so it is essential that this be done by well informed experts such as Roger Summons, Simon George and Jochen Brocks.

Comment from referee: "There is a need to discuss the possibility that there could have been contamination by organic compounds derived from the 2.7-2.8 Ga Fortescue Group that overlies the Dresser Fm in the studied area. It is conceivable that a "hydrothermal pump" could have circulated fluids downwards into the older succession. There is a significant literature on the organic geochemistry of the Fortescue Group by George, Coffey, Summons and others. The same applies to the Strelley Pool Fm where there are abundant stromatolites and microfossils".

Author's response: The depositional age of the Dresser Formation is constrained to ca. 3481 ± 3.6 Ma (Van Kranendonk et al., 2008). An emplacement of organic material into the Dresser hydrothermal chert vein by fluids circulating during the deposition of stratigraphically younger units as e.g. the Strelley Pool Formation (ca. 3.43–3.35 Ga; Hickman, 2008) and Fortescue Group (ca. 2.7–2.6, see reviewer comment) is unlikely. Firstly, there is no petrographic evidence that the analysed Dresser hydrothermal chert vein has been affected by fluid-flow events that post-date the initial formation (e.g. brecciation textures, etc.). The kerogen exclusively occurs in form of fluffy aggregates and clots embedded within a very dense chert matrix which is, once solidified, highly impermeable to fluids. It does furthermore not occur along grain boundaries, cracks etc. which could indicate possible transport by later fluids. This has also been described for other hydrothermal chert veins of the Dresser Formation, where the kerogen has been interpreted as being syngeneic (i.e. formed prior to host rock lithification; Morag et al., 2016). A later transport of kerogen without fluids, as alternative scenario, can also be excluded as macromolecular organic matter is not mobile in solid, impermeable materials.

Author's changes in manuscript: We tried to make this clearer (first paragraph of chapter 4.2 Syngeneity of the Dresser kerogen-derived compounds), now: "The kerogen of the Dresser Formation exclusively occurs in form of fluffy aggregates and clots embedded within a very dense chert matrix that is, once solidified, highly

impermeable to fluids. The depositional age of the formation is constrained to 3481 ± 3.6 Ma (Van Kranendonk et al., 2008), and the investigated chert vein shows no evidence for disruption by post-depositional hydrothermal fluids. This has also been described for other hydrothermal chert veins of the Dresser Formation, where the kerogen has been interpreted as being syngenetic (i.e. formed prior to host rock lithification; Ueno et al., 2001, 2004; Morag et al., 2016). Furthermore, the maturity of the embedded kerogen is in good accordance with the thermal history of the host rock. An introduction of solid macromolecular organic matter from stratigraphically younger units in this region during later fluid flow phases, as proposed for the younger Apex chert (Olcott-Marshall et al., 2014), can therefore be excluded".
* * *
Comment from referee: "The manuscript would be enhanced by adding a paragraph outlining the evidence that indicates that the 3.5Ga environment was anoxic".

Author's response & changes in manuscript: We now provide information on existing evidence for reducing conditions during deposition of the Dresser Formation (widespread presence of pyrite, Fe-rich carbonates, trace element signatures) and cite the relevant studies (Van Kranendonk et al., 2003, 2008) (chapter 4.3 Origin of the Dresser kerogen: hydrothermal vs. biological origin).
* * *
Comment from referee: "It seems to me that the evidence from the Apex Chert needs to be at least briefly reviewed here as it would add significantly to the context of this new work".

Author's response & changes in manuscript: In contrast to the Dresser hydrothermal chert vein analyzed in our study, the younger Apex chert has been affected by multiple fluid flow events. Some of these events significantly post-date the initial formation time and also led to an emplacement of younger organic materials (Olcott-Marshall et al., 2014). Our hypothesis may be relevant for the Apex chert in that it explains the

possible presence of organic matter during its initial formation. It cannot, however, help to pinpoint the formation pathways of distinct carbonaceous structures (e.g. Schopf, 1993, 2002; Brasier et al., 2002, 2005). We now explicitly state this problem in the manuscript (last paragraph of chapter 4.4 The "hydrothermal pump hypothesis").
* * *
Comment from referee: "The manuscript is well written and almost free of errors. P.11 l. 19 change to instantaneously".

Author's response & changes in manuscript: Done.
* * *
Comment from referee: "References to the published geological maps of the North Pole area should be added. There may be other publications by Hickman that should be cited". Author's response & changes in manuscript: We now provide information on published geological maps by Hickman (1983), Van Kranendonk (1999) and Hickman and Van Kranendonk (2012) (chapter 2.1). We also cite further publications by Hickman (1973, 1975, 2012) (chapters 1 Introduction and 4.1 Maturity of the kerogen).
* * *
References cited in the reply:

Brasier, M. D., Green, O. R., Jephcoat, A. P., Kleppe, A. K., Van Kranendonk, M. J., Lindsay, J. F., Steele, A., and Grassineau, N. V.: Questioning the evidence for Earth's oldest fossils, Nature, 416, 76–81, doi:10.1038/416076a, 2002.

Brasier, M. D., Green, O. R., Lindsay, J. F., McLoughlin, N., Steele, A., and Stoakes, C.: Critical testing of Earth's oldest putative fossil assemblage from the ∼3.5 Ga Apex chert, Chinaman Creek, Western Australia, Precambrian Res., 140, 55–102, doi:10.1016/j.precamres.2005.06.008, 2005.

Hickman, A. H.: The North Pole barite deposits, Pilbara Goldfield, Geol. Surv. West. Aust. Ann. Rep., 1972, 57–60, 1973.

Hickman, A. H.: Precambrian structural geology of part of the Pilbara region, Geol. Surv. West. Aust. Ann. Rep., 68–73, 1975.

Hickman, A. H.: Geology of the Pilbara block and its environs, Geol. Surv. West. Aust. Bull., 127, 1983.

Hickman, A. H.: Review of the Pilbara Craton and Fortescue Basin, Western Australia: crustal evolution providing environments for early life. Island Arc, 21, 1–31, 2012.

Hickman, A. H., and Van Kranendonk, M. J.: A Billion Years of Earth History: A Geological Transect Through the Pilbara Craton and the Mount Bruce Supergroup – a field guide to accompany 34th IGC Excursion WA-2, Geol. Surv. West. Aust., Record 2012/10, 2012.

Hickman, A. H.: Regional Review of the 3426–3350 Ma Strelley Pool Formation, Pilbara Craton, Western Australia. Geol. Surv. West. Aust., Record 2008/15, 2008.

Morag, N., Williford, K. H., Kitajima, K., Philippot, P., Van Kranendonk, M. J., Lepot, K., Thomazo, C., and Valley, J. W.: Microstructure-specific carbon isotopic signatures of organic matter from âĹij3.5 Ga cherts of the Pilbara Craton support a biologic origin, Precambrian Res., 275, 429–449, doi:10.1016/j.precamres.2016.01.014, 2016.

Olcott-Marshall, A., Jehlička, J., Rouzaud, J. N., and Marshall, C. P.: Multiple generations of carbonaceous material deposited in Apex chert by basin-scale pervasive hydrothermal fluid flow, Gondwana Res., 25, 284–289, doi: 10.1016/j.gr.2013.04.006, 2014.

Schopf, J. W.: Microfossils of the Early Archean Apex chert: new evidence of the antiquity of life, Science, 260, 640–646, doi: 10.1126/science.260.5108.640, 1993.

Schopf, J. W., Kudryavtsev, A. B., Agresti, D. G., Wdowiak, T. J., and Czaja, A. D.: Laser-Raman imagery of Earth's earliest fossils, Nature, 416, 73–76, doi:10.1038/416073a, 2002.

Van Kranendonk, M. J.: North Shaw, W.A. Sheet 2755: West. Aust. Geol. Surv. 1:100 000 Geol. Series, 1999.

Van Kranendonk, M. J., Webb, G. E., and Kamber, B. S.: Geological and trace element evidence for a marine sedimentary environment of deposition and biogenicity of 3.45 Ga stromatolitic carbonates in the Pilbara Craton, and support for a reducing Archaean ocean, Geobiology, 1, 91–108, doi:10.1046/j.1472-4669.2003.00014.x, 2003.

Van Kranendonk, M. J., Philippot, P., Lepot, K., Bodorkos, S., and Pirajno, F.: Geological setting of Earth's oldest fossils in the ca. 3.5 Ga Dresser Formation, Pilbara Craton, Western Australia. Precambrian Res., 167, 93–124, doi:10.1016/j.precamres.2008.07.003, 2008.

––––––––––––––––––––––

---

## Author Comment (AC2) · 5 Feb 2018

Comment from referee: "This is a very interesting paper showing a potential key characteristic of kerogen in chert veins of the early Archean Dresser Fm. The origin of the kerogen in this chert vein (and other early Archean chert veins in the Pilbara) still remains debated. It could represent sedimentary biomass that was recirculated by hydrothermal fluids, or represent the abiogenic product of FTT synthesis that was generated during hydrothermal serpentinization of ultramafic crust. This paper therefore addresses a very relevant scientific question. Based

on HyPy extraction, GC-MS and compound-specific isotope analysis, the authors showed that the kerogen phase releases n-alkanes with a specific carbon-number distribution. The sharp decrease in n-alkane abundance above n-C18, is similar to the distribution of n-alkanes released during HyPy of cyanobacteria. This distribution is not observed, however, in n-alkane products of FTT synthesis. Based on these observations, the authors conclude that the kerogen in the chert vein of the Dresser Fm is derived from biomass (such as e.g. phototrophs, chemolithoautrophs), and not from FTT synthesis associated with hydrothermal circulation. In order to explain the presence of kerogen in deep feeder chert veins, the authors propose a 'hydrothermal pump' hypothesis, in which redistribution and sequestration of microbial biomass occurs through hydrothermal circulation. This is a very nice explanation, and would confirm that microbial life was abundantly present in the ancient oceans. The paper is very well written, and experimental results support the conclusions. I have 3 issues that should be worked out better in this paper, which are listed below".
* * *
Comment from referee: "The conclusion of this paper strongly depends on the drop in >C18 n-alkanes in biologic materials. It should then be explained in detail why this happens. Is this drop observed in all biologic materials? Is it related to specific compounds that are present in cell membranes? There is now only a very short description about this (P10, line 10-11), stating that bacteria commonly (though not exclusively) form linear carbon chains <C18 (cf. Kaneda, 1991). I think it is important that this discussion is expanded here. Is there a chemical reason for this drop in abundance of n-alkanes beyond C18?

Author's response: Bacteria form acetyl-based hydrocarbon moieties such as fatty acids (the potential precursors of the kerogen-derived n-alkanes) as membrane- or storage lipids. Different biosynthetic pathways exist for the production of these lipids, which control carbon chain-lengths, number and positions of double bonds etc. These biosynthetic mechanisms typically result in the formation of lipids with chain-lengths ≤

n-C18.

Author's changes in manuscript: We now provide this information in the discussion (chapter 4.3 Origin of the Dresser kerogen: hydrothermal vs. biological origin), now: "[…] Straight-chain (acetyl-based) hydrocarbon moieties such as fatty acids – the potential precursors of the kerogen-derived n-alkanes – are to the current knowledge being formed only by Bacteria and Eukarya, where they typically function as constituents of membranes or storage lipids (cf. Kaneda, 1991; Erwin, 2012). The formation of these lipids is tightly controlled by different biosynthetic pathways resulting in characteristic chain-length distributions. In bacterial lipids, carbon chain-lengths typically do not extend above n-C18 (cf. Kaneda, 1991). […]".
* * *
Comment from referee: "The HyPy products of kerogen of the Dresser Fm are compared with HyPy products of cyanobacteria, but – if I understand correctly – not with the HyPy products of FTT synthesized carbonaceous solids. Only the direct gaseous FTT synthesis products are compared. Were there any solid phases produced during FTT synthesis? It would have been important to see what kind of HyPy products this would have generated, in comparison with that of HyPy products of cyanobacteria and Dresser Fm kerogen. I think the authors should discuss this better, in order to make clear that a comparison with a true abiogenic carbonaceous solid has not been made here".

Author's response: We agree that comparison is not equal. However, the presence of a true 'FTT-kerogen' (i.e. chemically equivalent to a biological kerogen in that it contains GC-amenable organic moieties) has not been observed in our experiments and has also not been reported in the literature. Conceivably, however, initially soluble functionalized FTT-products may evolve into such a 'FTT-kerogen' through diagenetic condensation reactions. It is quite certain that carbon chains released from this material would not show any preferences of distinctive homologues, but rather mirror the unimodal distribution of the initial educts.

Author's changes in manuscript: We included this aspect into the discussion (chapter 4.3 Origin of the Dresser kerogen: hydrothermal vs. biological origin), now: "[. . .]. In contrast, abiotically synthesised extractable organic compounds show a unimodal homologue distribution from the beginning (Fig. 3e) and will retain it, while thermal maturation would gradually shift the n-alkane pattern towards shorter homologues. It can therefore be expected that organic compounds cleaved from an abiotic 'Fischer-Tropsch-kerogen' – whose existence has not been proven yet – would also exhibit a unimodal distribution. [. . .]".
* * *
Comment from referee: "The maturity of kerogen in the Dresser Fm is determined using Raman spectroscopy. I have several comments to the methods used here. The H/C ratio of the kerogen fractions is calculated following the method of Ferralis et al. (2016), using a D5-peak in the Raman spectrum. However, based on the example spectrum in Fig.2c I think it is quite difficult to convincingly fit a D5-peak. The authors should at least show how this fit was made. It may well be that the kerogen has reached a degree of alteration where the amount of H is too low to create a significant D5-peak. The other method used here is to check the S1 peak (2450 cm-1) in the second order spectrum. The absence of this peak is consistent with kerogen that has experienced prehnite-pumpellyite to lowergreenschist-facies metamorphism. However, I think there are more precise methods to determine the degree of alteration of the kerogen. For instance, the Raman-based geothermometer of Koketsu et al. (2014) could have been applied to the first-order spectrum. Or even better, the Raman-based indicator of Delarue et al. (2016) can be applied. This indicator has been specifically developed to compare kerogens in Precrambrian cherts".

Author's response & changes in manuscript: The D5 peaks were fitted in the Lab-SpecTM software (version 5.19.17; Jobin-Yvon, Villeneuve d'Ascq, France) using the Gauss/Lorentz function. We now provide this information (chapter 2 Material & methods) and stress that Raman-based H/C-ratios for highly mature organic

matter should be treated with caution (chapter 4.1 Maturity of the Dresser kerogen). We now also applied further Raman-based maturity proxies and cite the relevant studies (Delarue et al., 2016) (chapter 4.1 Maturity of the Dresser kerogen).
* * *
Comment from referee: "P3, Line 27: : : :.(GC-MS). Before: : :."

Author's response & changes in manuscript: Done.
* * *
Comment from referee: "P4, Line 7 and line 22: Why is the heating program for the cyanobacteria different than for the extracted kerogens?"

Author's response & changes in manuscript: Archaean kerogens are easily contaminated. HyPy of Archaean kerogens therefore requires the addition of a thermal desorption step (∼350°C) prior to high temperature HyPy (550°C) to remove organic contaminants (cf. Brocks et al., 2003; Marshall et al., 2007). Recent biomass is much more reactive than (highly mature) fossil kerogens and therefore shows a distinctly different behavior during pyrolysis (Love et al. 2005). For this reason, HyPy of recent biomass necessitates a modified experimental protocol that increases reaction efficiencies and minimizes artefact formation. We now provide these information (chapter 2 Material & methods).
* * *
Comment from referee: "Section 2.8: It may be good for the reader if a little more information is given here on PAH's. What are they, and why are these ratios important? And how were these PAH's measured?"

Author's response & changes in manuscript: We now provide additional information and references (Killops and Killops, 2005; Peters et al., 2005) on the nature of polyaromatic hydrocarbons (PAH) and the utility of GC-amenable PAHs as maturity indicators (in chapter 2 Material & methods).

Comment from referee: "Section 4.4: The way it is written, it is not clear whether the 'hydrothermal pump hypothesis' is an already existing term, or is here proposed for the first time. It should be more clearly stated that this is a new proposed model."

Author's response & changes in manuscript: We clarified that we propose the 'hydrothermal pump hypothesis' within our study (chapter 4.4 The "hydrothermal pump hypothesis").
* * *
References cited in the reply:

Brocks, J. J., Love, G. D., Snape, C. E., Logan, G. A., Summons, R. E., and Buick, R.: Release of bound aromatic hydrocarbons from late Archean and Mesoproterozoic kerogens via hydropyrolysis, Geochim. Cosmochim. Acta, 67, 1521–1530, doi:10.1016/S0016-7037(02)01302-9, 2003.

Erwin, J. (Ed.): Lipids and biomembranes of eukaryotic microorganisms, Acad. Press, New York, U.S.A, 1973.

Kaneda, T.: Iso- and Anteiso-Fatty Acids in Bacteria: Biosynthesis, Function, and Taxonomic significance, Microbiol. Rev., 55, 288–302, 1991.

Killops, S. D., and Killops, V. J.: Introduction to organic geochemistry, Blackwell Publ., Malden, U.S.A., 2005.

Love, G. D., Bowden, S. A., Jahnke, L. L., Snape, C. E., Campbell, C. N., Day, J. G., and Summons, R. E.: A catalytic hydropyrolysis method for the rapid screening of microbial cultures for lipid biomarkers, Org. Geochem., 36, 63–82, doi:10.1016/j.orggeochem.2004.07.010, 2005.

Marshall, C. P., Love, G. D., Snape, C. E., Hill, A. C., Allwood, A. C., Walter, M. R., Van Kranendonk, M. J., Bowden, S. A., Sylva, S. P., Summons, R. E.: Structural character-

ization of kerogen in 3.4 Ga Archaean cherts from the Pilbara Craton, Western Australia, Precambrian Res., 155, 1, 1–23, doi:10.1016/j.precamres.2006.12.014, 2007, 2007.

Peters, K. E., Walters, C. C., and Moldowan, J. M.: The Biomarker Guide: Volume 2, Biomarkers and Isotopes in Petroleum Exploration and Earth History, Cambridge University Press, Cambridge

---

## Author Response (AR2)

**GEORG–AUGUST-UNIVERSITY OF GOETTINGEN**

[Figure]

Centre of Geoscience
**Division of Geobiology**
*Dr Jan-Peter Duda*

GZG▪Univ.Göttingen▪Geobiologie▪Goldschmidtstr.3▪37077 Göttingen▪Germany

**Associate editor Biogeosciences**

Prof. Dr. Jack Middelburg

- - -

Goldschmidtstr. 3
37077 Göttingen
Germany
Phone:  +49(0)551-39 7960
Facsimile: +49(0)551-39 7918
E-mail: jduda@gwdg.de
http://www.
geobiologie.uni-goettingen.de

[Figure]

Göttingen, 09/02/2018

**Submission of revised manuscript (after final technical corrections)**

Dear Prof. Middelburg,

I would once again like to thank you and the reviewers for your support and helpful comments. We now completed the requested final corrections and appreciate your efforts in the acknowledgments.

We trust that the revised manuscript meets the requirements. Thank you again!

Yours sincerely,

Jan-Peter Duda (on behalf of the co-authors)